# Dual Role of the PTPN13 Tyrosine Phosphatase in Cancer

**DOI:** 10.3390/biom10121659

**Published:** 2020-12-11

**Authors:** Soha Mcheik, Leticia Aptecar, Peter Coopman, Véronique D’Hondt, Gilles Freiss

**Affiliations:** 1IRCM, Institut de Recherche en Cancérologie de Montpellier, INSERM U1194, Université de Montpellier, Institut régional du Cancer de Montpellier, F-34298 Montpellier, France; soha.mcheik@inserm.fr (S.M.); l.aptecar@hotmail.fr (L.A.); peter.coopman@inserm.fr (P.C.); Veronique.Dhondt@icm.unicancer.fr (V.D.); 2CNRS—Centre National de la Recherche Scientifique, 1919 Route de Mende, 34293 Montpellier, France

**Keywords:** PTPN13, cell signaling, cancer, tumor suppressor, oncogene

## Abstract

In this review article, we present the current knowledge on PTPN13, a class I non-receptor protein tyrosine phosphatase identified in 1994. We focus particularly on its role in cancer, where PTPN13 acts as an oncogenic protein and also a tumor suppressor. To try to understand these apparent contradictory functions, we discuss PTPN13 implication in the FAS and oncogenic tyrosine kinase signaling pathways and in the associated biological activities, as well as its post-transcriptional and epigenetic regulation. Then, we describe PTPN13 clinical significance as a prognostic marker in different cancer types and its impact on anti-cancer treatment sensitivity. Finally, we present future research axes following recent findings on its role in cell junction regulation that implicate PTPN13 in cell death and cell migration, two major hallmarks of tumor formation and progression.

## 1. Introduction

Tyrosine phosphorylation is a key post-translational modification and a major research topic because of its crucial role in the control of cell proliferation, cycle progression, differentiation, and development. The interest in protein tyrosine phosphatases has been growing steadily in the last 25 years.

In this review, we focus on protein tyrosine phosphatase non-receptor 13 (PTPN13) because many studies have highlighted its possible dual role in cancer. PTPN13 was first named PTP-BAS [1], hPTP1E [2], and PTP-L1 [3] since it was cloned by three different groups, then renamed FAS-associated phosphatase 1 (FAP-1) by Sato et al because of its interaction with Fas [4]. PTPN13 belongs to the Class I superfamily of tyrosine-specific phosphatases in which the catalytic domain contains a cysteine residue [5]. Substitution of this cysteine residue by a serine results in a catalytically inactive form (PTPN13-C/S) [6]. PTPN13 is a soluble and cytosolic non-receptor protein that can be translocated to the sub-membranous and nuclear compartments [7] during mitosis [8]. The PTPN13 gene is located on chromosome 4q21.3 and encodes the non-receptor protein phosphatase with the highest molecular weight (270 kDa; 2466 amino acids) [9].

PTPN13 is composed of an N-terminal KIND domain of unknown function, followed by a FERM domain and five PDZ domains before the catalytic phosphatase domain located at the C terminus [1] (Figure 1). The FERM domain is common to the family of membrane proteins that link the cytoskeleton to the plasma membrane [10]. The FERM domain of PTP-BL (PTPN13 murine ortholog with 80% of homology [11]) is sufficient to support its sub-membranous localization [12]. We showed that PTPN13 is mainly localized at the plasma membrane in HeLa cells, and that the FERM domain is necessary and sufficient to direct the enzyme to the membrane [13]. Kimber et al. showed that tandem pleckstrin homology-domain-containing protein (TAPP1), which binds PtdIns(3,4)P2 and the first PDZ domain of PTPN13, participates in the regulation of its membrane localization [14]. Moreover, the FERM domain allows the interaction of PTPN13 with other proteins, such as serologically defined colon cancer antigen 3/endosome-associated trafficking regulator 1 (SDCCAG3/ENTR1) that is overexpressed in colon cancer and is involved in cytokinesis regulation [15]. PDZ domains are protein interaction domains that are used as scaffolding platforms in protein complexes, mostly associated with the plasma membrane [16]. Multiple partners of PTPN13 PDZ domains have been identified [17] and they are often involved in carcinogenesis and/or in cytoskeleton organization and cell migration (Figure 1; Table 1).

Little is known about PTPN13 physiological role. Ptpn13 gene ablation does not induce major changes in mice. Its loss in murine CD4 T cells increases their differentiation into Th1 and Th2 helper T lymphocytes and enhances the immune defenses against *Klebsiella pneumoniae* [33]. Moreover, PTP-BL plays a role in adipocyte differentiation [34], and loss of its catalytic activity slightly affects motor neuron repair in mice [35]. In this review, we mainly describe its implication in cancer-related signaling pathways and biological processes (Figure 2, Table 2).

## 2. Signaling Pathways and Biological Activities Affected by PTPN13

### 2.1. FAS Pathway

PTPN13 role in FAS-mediated apoptosis has been extensively studied. The first reported evidence was the interaction of PTPN13 with the FAS (CD95/APO-1) death receptor. Many cancer cell lines develop resistance to FAS-induced apoptosis by acquiring mutations in the FAS gene [44] or by regulating FAS availability at the cell surface [45]. It has been demonstrated that PTPN13 reduces FAS cell surface level, and that the second and fourth PDZ domain of PTPN13 directly interact with FAS C-terminus [27,46,47]. Moreover, blocking this PDZ domain with a serine/leucine/valine (SLV) tripeptide, which represents the FAS C-terminus sequence required for the association with PTPN13, prevents the interaction of PTPN13 with partner proteins, leading to restoration of FAS-induced apoptosis [47,48,49,50,51]. In agreement with these findings, Gagnoux-Palacios L et al. showed that FAS interacts via its SLV terminal sequence with PDZ domain-containing proteins associated with cell junctions [52], such as the scaffolding protein Discs Large homolog 1 (DLG1) and PTPN13 that is involved in cell junction stabilization in breast cancer [53]. They then demonstrated that adherens junctions (AJs) play a role in tissue homeostasis by sequestering FAS and inhibiting apoptosis. AJ disruption allows the release of FAS and increases the number of cells sensitive to FasL-mediated cell death [52]. They also established that transfection of HCT15 cells with siRNAs against DLG1 increases FAS-induced cell death. As this effect is correlated with an increase in total and cell surface FAS expression, DLG1 might inhibit FAS cell death by impairing Death Inducing Signaling Complex (DISC) formation [52].

Using siRNA-mediated knockdown or overexpression, Ivanov et al. demonstrated PTPN13 implication in FAS localization. Specifically, forced expression of PTPN13 in FEMX melanoma and HeLa cervical carcinoma cells leads to reduced surface and higher intracellular expression of FAS. Conversely, FAS surface expression is upregulated after transfection of TIG3 bladder carcinoma cells with PTPN13 dominant negative mutants or after PTPN13 silencing by siRNA [45]. In line with these findings, PTPN13 overexpression in Capan-1 pancreatic carcinoma cells negatively regulate FAS-mediated apoptosis [54], and PTPN13 silencing by siRNA in SW480 colorectal cancer cells increases FAS/FasL-mediated apoptosis [55]. Consistent with these results, another study reported that the microRNA mir-200c sensitizes cells to FAS-induced apoptosis by targeting PTPN13 [56].

Erdmann’s group proposed a mechanism whereby PTPN13 regulates FAS membrane trafficking. First, they showed that in HeLa cells, PTPN13 can bind to ENTR1 (an endosome-associated trafficking regulator that controls sensitivity to tumor necrosis factor-induced apoptosis [57], also known as SDCCAG3) and that these two partners play a role in cytokinesis [15]. Then, they confirmed that ENTR1 negatively regulates FAS surface expression and FAS-induced apoptosis by participating in FAS endolysosomal sorting [58]. In addition, they reported that the regulation of FAS surface expression requires the interaction between PTPN13 and ENTR1, and the colocalization of FAS, ENTR1, and PTPN13 in early endosomes [58].

Eklund’s group showed that in myeloid progenitor cells, BCR/ABL activation leads to PTPN13 upregulation and inhibition of FAS-induced apoptosis. This inhibitory effect is abolished by the SLV peptide [59], suggesting PTPN13 involvement. Interestingly, another study reported that PTPN13 can dephosphorylate and inhibit c-Abl[30], suggesting a possible retro-control of ABL activity that is lost in the case of the BCR/ABL translocation. Then, Eklund’s group demonstrated that in BCR/ABL-overexpressing myeloid progenitor cells, PTPN13 interacts with adenomatous polyposis coli (APC), and that this interaction, which involves the same PDZ domain, is inhibited by the SLV peptide [60]. BCR/ABL overexpression leads to a decrease in tyrosine phosphorylation of GSK3β (an APC partner) and in serine phosphorylation of β-catenin (a GSK3 substrate), and consequently to increased β-catenin transcriptional activity [60]. All these BCR/ABL-mediated effects can be reversed by the SLV peptide, suggesting PTPN13 implication. Using a murine model of chronic myeloid leukemia (CML) stem cells transplantation, Eklund’s group reported that administration of the SLV peptide (to inhibit PTPN13), in addition to the standard treatment with the BCR/ABL inhibitor imatinib, delays tumor development, suggesting an inhibition of leukemia stem cell persistence [61]. More recently, they showed that increased PTPN13 expression characterizes CD133+ colon cancer stem cells and that PTPN13 upregulation in metastases, compared with primary tumors or after platinum-based chemotherapy, is due to the relative abundance of these cancer stem cells in the tumor [51]. 

In line with these results, Sardina et al. observed increased PTPN13 expression during megakaryocytic differentiation. This inhibits differentiation and limits tyrosine and serine phosphorylation of β-catenin, leading to higher transcriptional activity [43]. In this model, PTPN13 overexpression is induced by the Wnt pathway that stabilizes PTPN13, and the PTPN13-β-catenin interaction appears to be direct [43]. 

On the other hand, Castilla et al. found that PTPN13 plays a pro-apoptotic role in PC3 and LNCaP prostate adenocarcinoma cells. Specifically, its overexpression enhances apoptosis in cells incubated with phenylethyl isothiocyanate and anti-FAS antibodies or with paclitaxel [43]. Conversely, PTPN13 downregulation leads to resistance to apoptosis upon exposure to these agents [62]. Furthermore, they showed that PKCδ acts as an intermediary in PTPN13-mediated apoptotic signaling, and that inhibition of IκBα degradation and suppression of NF-κB activity by IκBα association and dephosphorylation are partly regulated by PTPN13 and PKCδ. Finally, they reported that PTPN13 and PKCδ expression are lost in poorly differentiated, more aggressive human prostate cancer specimens, suggesting a correlation between their absence, apoptosis resistance and tumor progression [62].

In the pancreatic adenocarcinoma A818-6 cell line, anti-FAS antibodies induce apoptosis in cells grown in 2D- or 3D-polarized cell cultures, independently of PTPN13 and FAS colocalization [63]. Moreover, inhibition of PTPN13 expression does not affect apoptosis induced by anti-FAS agonistic antibodies, suggesting that PTPN13 is not involved in the regulation of FAS-induced apoptosis in this cell type [63].

In conclusion, PTPN13 inhibitory role in FAS-induced apoptosis is well documented in the hematopoietic system. Conversely, its role in solid tumors is less clear due to the multiple signaling pathways regulated by PTPN13, and particularly due to its role in cell junction maintenance that contributes to FAS-induced apoptosis regulation.

### 2.2. SRC, Ephrin, ErbB Pathways

SRC is a non-receptor pro-oncogenic tyrosine kinase that participates in different pathways, for instance, the integrin [64], EGFR [65], and ephrin signaling cascades [40,66].

SRC [38], ephrinB [40], EGFR [67], and HER2 [37,38] signaling are inhibited by PTPN13 that limits anchorage-independent cell growth (SRC) [49], cell proliferation (HER2 and EGFR) [67], and also invasion and tumor aggressiveness (SRC, HER2) [37,38].

Several PTPN13 partners involved in these inhibitory effects have been identified. For instance, in HCT116 colon cancer cells, PTPN13 inactivates SRC through interaction with reversion-induced LIM domain protein (RIL) [68], and in A549 pulmonary and Caco-2 colon adenocarcinoma cells, it inhibits the ERBB3/ERBB2 receptors through association with nectin-like molecule 2 (NECL2) [18]. In addition, there may be a HER2-induced feedback mechanism that induces the expression of PTPN13 and thereby negatively regulates its own signaling pathways [69].

In 2012, Vermeer et al. discovered a new signaling pathway that involves ephrinB1, ERBB2, and ERK and that is regulated by PTPN13 in breast cancer cell lines [70]. Specifically, PTPN13 silencing by shRNA leads to ERBB2 overexpression and to phosphorylation of ephrinB1 (a PTPN13 substrate) that forms a complex with ERBB2 and induces ERK1/2 phosphorylation in breast cancer (MDA-MB-468), human keratinocyte (HaCaT) and head and neck squamous cancer (UM-SCC84) cell lines. Formation of the ephrinB1/ERBB2 complex was confirmed by coimmunoprecipitation experiments and by their colocalization at intercellular junctions. Moreover, ephrinB1/ERBB2 complex formation is promoted by expression of a constitutively active oncogenic mutant of ERBB2. Signaling by this complex is activated by SRC and is increased by transfection of the catalytically inactive PTPN13-C/S variant. These findings were also confirmed by proximity ligation assay and ephrinB1/ERBB2 coimmunoprecipitation in head and neck cancer cell lines [71]. Besides ERBB2, ephrinB1 interacts with ERBB1 that is overexpressed in head and neck cancer [72]. This interaction, which is promoted by human papilloma virus 16 (HPV16) (known to inhibit PTPN13 in this cancer type [73]), activates the MAPK pathway [70]. Transfection of HEK293 cells with the PTPN13-C/S mutant increases ephrinB1 phosphorylation, its association with ERBB1, and ERK1/2 phosphorylation. These effects are not observed upon transfection of wild-type PTPN13 [71]. Altogether, these data indicate that PTPN13 regulates phosphorylation of ephrinB1, which preferentially associates with ERBB1 in head and neck cancer, and allows MAPK pathway activation.

In vivo, in a mouse model of HPV-positive head and neck squamous cancer (HNSCC), tumor growth was reduced (*p* < 0.001) and survival was improved (*p* < 0.001) in mice injected with HNSCC cells transfected with ephrinB1 shRNA compared with cells transfected wild-type ephrinB1 [71]. This suggests that ephrinB1 may have a pro-oncogenic role through activation of the MAPK pathway via the ephrinB1/ERBB2 [70] or ephrinB1/ERBB1 complexes [71].

Other studies on PTPN13 regulation using miRNAs support these results, notably in bronchial adenocarcinoma [74,75] and esophageal carcinoma [76] cell lines. Moreover, NECL2 and NECL4 can interact with PTPN13 to inhibit the ERBB2/ERBB3 pathway in colorectal cancer (Caco-2), breast cancer (MCF7), and human embryonic kidney (HEK293) cells [77,78].

In conclusion, the ephrinB1/ERBB2 (in breast cancer) and ephrinB1/ERBB1 (in head and neck cancer) complexes induce MAPK pathway activation and are positively regulated by SRC and negatively regulated by PTPN13.

### 2.3. PTPN13 Is Implicated in HPV16 Carcinogenicity

The HPV proteins E6 and E7 exert their oncogenic activity through inhibition of the p53 and pRb tumor suppressor genes, respectively [79,80]. In addition, E6 binds to several PDZ domain-containing proteins, including PTPN13, via a PDZ binding domain (PBM), leading to their proteasome-mediated degradation [81,82]. By transfecting wild-type and PBM-mutated E6 proteins in human and mouse keratinocytes, Spanos’ group confirmed that HPV16-induced degradation of PTPN13 promotes anchorage-independent tumor cell proliferation [59]. In vivo, expression of an activated RAS mutant enhances the growth of epithelial cells (MTECs) transfected with wild-type E6 [83]. This RAS effect is not observed when cells are co-transfected with PTPN13 shRNA or PBM-mutated E6, suggesting that RAS activation acts in conjunction with PTPN13 inhibition to promote tumor proliferation.

More recently, Wieking et al. showed that HPV inhibition of PTPN13 and p53 is involved also in the induction of epithelial mesenchymal transition (EMT). Indeed, infection of human tonsillar epithelial cells with a non-oncogenic mutated HPV16 virus (unable to degrade p53, pRb, PTPN13, or to activate telomerases) did not induce EMT, unlike transfection with wild-type HPV16 [84].

Thus, E6-mediated inhibition of PTPN13 contributes to E6 oncogenicity by promoting anchorage-independent growth and EMT in virus-infected cells.

### 2.4. NF-κB Pathway

After exposure to oxidative stress or hypoxia, IκBα phosphorylation at tyrosine 42 (Y42) releases NF-κB that activates transcription of genes involved in apoptosis resistance, cell proliferation, and immune and inflammatory responses [85,86,87,88]. Nakai et al. demonstrated that in vitro, IκBα is dephosphorylated at Y42 by PTPN13 [42]. These results were confirmed by Wang et al. in a high-grade serous ovarian carcinoma (HGSOC) cell line (OV-90), in which PTPN13 transfection decreased the levels of IκBα phosphorylated at Y42 and of nuclear NF-κB, in contrast to transfection with PTPN13 siRNA. By using a siRNA against PTPN13 in combination with an inhibitor of IκBα phosphorylation or an IκBα mutant (Y42A), they confirmed that PTPN13 exerts its tumor suppressive effect by dephosphorylating IκBα at Tyr42 [89]. These results could provide a mechanistic explanation for the work by Castilla et al. showing that PTPN13 and PKC-δ participate in NF-κB activation in PC3 prostate cancer cells [62] (see Section 2.1).

### 2.5. EMT, Cell Migration and Invasion

Using a PTPN13-transfected hepatocellular carcinoma (HCC) cell line, Zhan et al. suggested that PTPN13 negatively regulates EMT by inhibiting Slug and Snail, two master EMT transcription factors [90]. However, PTPN13 might hinder EMT also through cell junction stabilization via its positive role on desmosome formation, as demonstrated in vitro in MDA-MB-231 breast cancer cells that overexpress PTPN13 and in vivo in a transgenic mouse model that lacks PTPN13 [53]. Although inhibition of miRNA-200c, which targets PTPN13, is associated with EMT [33], an in vivo study found no significant association between downregulation of the miRNA-200 family, PTPN13 expression, and colorectal cancer metastatic potential [91]. 

PTPN13 is also involved in the regulation of cancer cell migration/invasion via its partner and substrate thyroid hormone receptor interactor 6 (TRIP6, also called ZRP-1) that promotes cell mobility induced by lysophosphatidic acid [41], and activates Wnt/β-catenin signaling [92]. 

Furthermore, siRNA-mediated PTPN13 silencing in the PC3 and DU145 prostate cancer cell lines leads to overexpression of urokinase-type plasminogen activator (uPA) fibrinolytic system components that can degrade the extracellular matrix, thus promoting tumor invasion [93].

In addition, two recent studies illustrate PTPN13 involvement in cell junction maintenance. Fan et al. [22] demonstrated that PTPN13 can dephosphorylate junctional adhesion molecule-A (JAM-A), a transmembrane component of tight junctions, leading to their stabilization. Then, analysis of the interactome of different cell junctions showed that in epithelial cells, PTPN13 is part of the apical marginal zone and interacts with tight junctions and cytoskeleton molecules [94].

In the hematopoietic system, PTPN13 silencing promotes hematopoietic stem cell (HSC) adhesion to bone marrow (thus decreasing their invasiveness), and increases their quiescence [95]. This study, based on previous results by Sardina et al. [43] showing that PTPN13 stabilizes β-catenin in megakaryocytes, found that inhibition of PTPN13 or β-catenin in vivo increases HSC adhesion to their niche. The authors hypothesized a negative transcriptional regulation of cell adhesion molecules by β-catenin, which is positively regulated by PTPN13 [95]. 

Thus, in solid cancers, PTPN13 inhibits primary tumor cell invasiveness through inhibition of the uPA system, regulation of the main EMT genes, and stabilization of cell junctions. In hematopoietic malignancies, PTPN13 promotes, through β-catenin, HSC adhesion to their niche, which may later lead to tumor cell invasion.

### 2.6. PI3K/PTEN Pathway

The PI3K pathway is often overactivated in different cancer types, particularly in breast cancer. PTPN13 effect on PI3K activation varies depending on the cell type. In human fibroblasts, HeLa and HEK293T cells, the results obtained by Kuchay et al. suggest that PTPN13 dephosphorylates the p85 regulatory subunit of PI3K that can then interact with F-box and leucine rich repeat protein (FBXL2). This leads to its ubiquitylation and degradation by the proteasome. The negative regulation of the p85 subunit by PTPN13 is important to maintain the insulin receptor substrate-1 (IRS-1)-mediated activation of the PI3K pathway [96].

Conversely, our group demonstrated that PTPN13 inhibits PI3K in MCF7 breast cancer cells [36,97]. PTPN13 is upregulated by antiestrogen agents and is required for apoptosis induction by this treatment [98]. Specifically, in PTPN13-expressing MCF7 cells, incubation with the antiestrogen agent tamoxifen severely reduces IRS-1 and Akt phosphorylation induced by IGF-1 and leads to a strong increase in apoptosis. These effects are abolished in cells transfected with PTPN13 antisense, confirming that PTPN13 promotes apoptosis by inhibiting the PI3K survival pathway [97]. In addition, we showed by in vitro and in cellulo substrate trapping, dephosphorylation, and colocalization experiments, that PTPN13 specifically dephosphorylates IRS-1. Very recently, it was demonstrated that PTPN13 also acts as a tumor suppressor in clear cell renal cell carcinoma (ccRCC) where lower PTPN13 expression levels predict shorter survival rate. Moreover, in ccRCC cell lines and xenografts, PTPN13 overexpression restricts cancer cell proliferation and invasion through Akt inactivation [99].

PTPN13 can also regulate the PI3K signaling pathway independently of IRS-1 by interacting with Phosphatase and TENsin homolog (PTEN). Yeast two-hybrid and GST pull-down assays showed that the second PDZ domain of PTPN13 can bind to PTEN [28]. Bruurs et al. later demonstrated that this interaction allows the apical localization of PTEN, resulting in the restriction of the apical membrane size in a colorectal cancer cell line. These effects were not modified by inhibition of PTPN13 catalytic activity, suggesting that in this context, PTPN13 functions as an anchor protein. Conversely, PTEN catalytic activity was still required [100].

## 3. Transcriptionnal, Post-Transcriptional, Genetic, and Epigenetic Regulation of PTPN13

PTPN13 expression and activity can be regulated during transcription, via methylation of the CpG islands within its promoter, after transcription by microRNAs or alternative splicing, and also after translation by specific signaling pathways.

### 3.1. Transcriptional Regulation 

#### 3.1.1. Transcriptional Regulation Mediated by Transcription Factors

Chromatin immunoprecipitation of the PTPN13 promoter revealed that STAT3, HDAC5 [101], and SMYD2 [102] are involved in its transcriptional regulation. The pro-tumor transcription factor STAT3, in combination with the nuclear co-repressor HDAC5, inhibits PTPN13 transcription in squamous cell lung carcinoma cell lines (HCC-1588 and SK-MES-1) after stimulation by the pro-tumor interleukin 6 [101]. The transcription factor SMYD2 is a negative regulator of PTPN13 transcription in polycystic kidney disease [103]. It also promotes tumorigenesis in mice bearing xenografts of triple negative breast cancer cells (MDA-MB231 and MDA-MB468). Similar results were obtained upon PTPN13 silencing by siRNA [102].

More recently, Yan Y. et al. demonstrated that, in HCC, the hepatitis B virus X protein regulates PTPN13 expression via the DNA methyltransferase 3A, which binds to the PTPN13 promoter and induces the hypermethylation of its CpG islands. This loss of PTPN13 leads to an increase in c-Myc levels and signaling through the loss of its competitive interaction with IGFP2B1 that protects c-Myc mRNA from degradation [104].

On the other hand, the interferon consensus sequence binding protein (ICSBP) transcription factor is a tumor suppressor involved in myelopoiesis, and its expression is repressed in CML [105]. Eklund’s group reported that ICSBP negatively regulates PTPN13 in CML cell lines, promoting FAS-mediated apoptosis [106]. They then showed that formation of the Tel-ICSBP-HDAC3 multiprotein complex is required for ICSBP-induced PTPN13 repression. In addition, the Tel-PGFRβ fusion protein, which results from a chromosomal translocation associated with leukemia, inhibits binding of the Tel/ICSBP/HDAC3 complex to the PTPN13 promoter. This restores PTPN13 expression, causing inhibition of FAS-induced apoptosis in CML [107].

To date, Ewing sarcoma protein-Friend leukemia integration 1 (EWS-FLI1) and homeobox C8 (Hox-C8) are the only known positive regulators of PTPN13 expression. In Ewing’s sarcomas, the t(11;22) translocation leads to the expression of the EWS-FLI1 fusion protein, an oncogenic transcription factor that activates PTPN13 transcription, inducing tumor proliferation [108]. The authors then reported a drastic decrease in cell survival after transfection of the PTPN13-C/S mutant in Ewing sarcoma cells. Using substrate trapping in vitro and in vivo in the human Ewing sarcoma TC32 cell line, they identified valosin-containing protein (VCP/p97) as a new PTPN13 substrate. As VCP/p97 phosphorylation is necessary for PTPN13 midbody localization during cell division, a PTPN13 pro-oncogenic role in Ewing sarcoma, mediated through regulation of cell division, could be suggested [39].

Hox-C8 is a transcription factor involved in cell differentiation and tissue migration [109]. PTPN13 is one of the genes commonly co-expressed with Hox-C8, and chromatin immunoprecipitation assays confirmed that Hox-C8 binds to the PTPN13 promoter. Moreover, Hox-C8 and PTPN13 expression levels are positively correlated. Hox-C8 overexpression in MC3T3-E1 immature osteoblasts, C3H10T1/2 mesenchymal stem cells, and NIH3T3 fibroblasts positively regulates PTPN13 expression levels, while Hox-C8 knockdown decreases them [109].

Unlike hematopoietic cancers where PTPN13 negative transcriptional regulation has an anti-tumor effect by sensitizing cells to FAS-induced apoptosis, PTPN13 negative transcriptional regulation in solid cancers has a pro-tumor effect through activation of proliferation pathways (Table 3).

#### 3.1.2. Transcriptional Regulation Mediated by PTPN13 Promoter Methylation

The PTPN13 and MAPK10 genes share a bi-directional promoter [110] that contains 12 CpG islands [111]. Its methylation is consistently associated with decreased PTPN13 expression [103], and has been observed in many hematologic (94% of 16 non-Hodgkin lymphoma cell lines, 50% of 6 Hodgkin lymphoma cell lines) and solid cancer cell lines (67% of 12 HCC cell lines, 60% of 10 gastric cancer cell lines, and 30% of 10 breast cancer cell lines). Similarly, Yeh et al. reported that the PTPN13 promoter is methylated in 66% of 12 HCC samples without loss of heterozygosity (LOH) of chromosome 4q [104]. Moreover, Wang et al. found that the PTPN13 promoter is methylated in 60% of 47 diffuse large B cell lymphoma samples, compared with 6.3% of 16 non-tumor tissue samples [112].

A recent study [62] established four esophageal and Barrett’s esophagus adenocarcinoma subtypes based on their genome methylation rate. PTPN13 was among the 69 genes in the highly methylated subtype. Specifically, it was methylated in 56% of 16 esophageal adenocarcinoma samples, and this was associated with decreased mRNA levels in 75% of cases. Conversely, the other subtypes did not show any PTPN13 methylation. Moreover, in vitro, shRNA-mediated PTPN13 silencing significantly increases proliferation and migration of SK-GT-4 esophageal adenocarcinoma cells. 

In all studies on PTPN13 promoter methylation, PTPN13 expression and activity were restored by incubating cells with 5-azacitidine, a DNA demethylating agent [76,112,113].

### 3.2. Post-Transcriptional Regulation 

#### 3.2.1. Post-Transcriptional Regulation by microRNAs 

MicroRNAs are non-coding RNAs that regulate gene expression through degradation or translation inhibition of their target.

The miR-30 family is composed of five members, a/b/c/d/e (for review: [114]). In tumor samples, miR30-e is decreased in bladder [115], breast [116], and rectal cancer [117], while it is overexpressed in salivary gland cancer [118] and pulmonary adenocarcinoma [119]. Zhuang’s laboratory confirmed that miR30-e is significantly overexpressed in lung adenocarcinoma compared with healthy tissue samples. They also showed that PTPN13 is a direct target of miR30-e. Tumor growth is promoted in mice xenografted with A549 cells transfected with miR30-e compared with cells transfected with vector alone. Moreover, while PTPN13 overexpression can reverse miR30-e effects on A549 cell growth, siRNA-mediated PTPN13 silencing enhances cell proliferation. This indicates that miR30-e effects are mediated through PTPN13 inhibition [75].

The miR-26 family includes three subtypes with altered expression in various tumors. Xu et al. showed that PTPN13 is a direct target of miR-26a in bronchial adenocarcinoma. Indeed, miR-26a overexpression leads to wild-type PTPN13 protein level reduction, but has no effect on the expression of a PTPN13 variant harboring a mutation in the putative miR-26a binding site. In the SPCA-1 lung adenocarcinoma cell line, siRNA-mediated PTPN13 knockdown mimics the effect of miR26-a, promoting phosphorylation of SRC, Akt, and ERK [74], supporting miR26a oncogenic role in bronchial adenocarcinoma.

The miR-200 family contains five subtypes (miR-200a/b/c, miR-141, and miR-429), and is involved in maintaining the epithelial phenotype [120]. Schickel et al. demonstrated that miR-200c decreases PTPN13 transcription by 60% in HEK293 cells. Furthermore, PTPN13 repression mediated by miR-200 increases sensitivity to FAS-induced apoptosis in tumor cell lines with mesenchymal features [56] (see chapter A).

The miR-185 family is overexpressed in bladder and kidney cancers and targets the PTPN13 and PTEN genes. However, their effects have not been studied in vivo yet [121] (Table 4).

#### 3.2.2. Post-Transcriptional Regulation by Alternative Splicing

Although there are four PTPN13 isoforms [122], the regulation and consequences of PTPN13 alternative splicing have been rarely explored. In the only published study, the consequences of hypoxia on the genome of prostate adenocarcinoma cells were analyzed by next generation sequencing. They found a decrease of more than 25% in PTPN13 exon inclusion rate, but the functions of its spliced isoforms remain unknown [123].

### 3.3. Post-Translational Regulation

Similarly, PTPN13 post-translational regulation remains largely unexplored. To our knowledge, only one study has been published. Wang et al found that in a mouse brain after traumatic brain injury, calpain-2 inhibits PTPN13 activity by upstream cleavage of its catalytic domain, leading to ABL-mediated phosphorylation of the microtubule-associated tau protein [30]. 

It would be of particular interest to study the role of the large number of PTPN13 post-translationally modified residues found in global proteomic studies, as illustrated in the www.phosphosite.org database, but their function and significance remain to be elucidated.

## 4. Medical Implication of PTPN13

### 4.1. Prognostic Marker of Survival

Several retrospective clinical studies analyzed PTPN13 prognostic value in different diseases.

In ovarian cancer, a first study of 95 specimens with different histological subtypes found no association between PTPN13 expression and survival at 2, 3, and 5 years [124]. However, two studies only on HGSOC, the most frequent ovarian cancer subtype, highlighted a correlation between high PTPN13 protein and mRNA expression and better prognosis in 97 and 58 HGSOC samples, respectively (*p* = 0.042 and *p* = 0.03) [89,125]. They also observed that PTPN13 expression levels are reduced in tumor tissues compared with normal tissues.

In breast cancer (n = 291 samples), we showed that high PTPN13 expression, measured by RT/PCR, is associated with better prognosis (*p* = 0.01 and RR = 0.48 in multivariate analysis) [126]. In a second study, in which we analyzed 24 breast cancer samples by immunohistochemistry (IHC), we observed a progressive decrease in PTPN13 expression from normal to metastatic tissue samples [38].

In prostate cancer samples (n = 76), PTPN13 expression, estimated by IHC, was inversely correlated with the Gleason score (*p* < 0.05). PTPN13 was overexpressed in well-differentiated tumors (low Gleason score), and downregulated in high-grade prostate tumors [62].

Three of four studies on lung cancer [squamous lung cell carcinoma (LSCC) [127], adenocarcinoma [75], and non-small-cell lung carcinoma (NSCLC) [67]] found a significant decrease in PTPN13 expression in tumors compared with normal tissues. Two of these studies also evaluated PTPN13 prognostic value, and found that RNA expression is associated with improved survival in 27 patients (HR = 0.28 *p* = 0.02) [128], and its protein expression with a lower aggressiveness in 91 primary LSCC samples (negative correlation with size, grade, and lymph node metastases, *p* < 0.001 for all three criteria) [127]. The fourth study [128] did not find any significant difference in PTPN13 expression between adenocarcinoma samples from non-smoking patients and adjacent normal tissues. 

A study on 282 HCC samples demonstrated that elevated PTPN13 expression (by IHC) is associated with better prognosis (*p* = 0.034) [90].

In primary Ewing sarcoma (n = 144), PTPN13 was detected by IHC in 80% of tumor specimens, but its level was not associated with survival [129]. Interestingly, it was not reported in normal human bone samples. 

In glioblastoma, PTPN13 RNA level was increased compared with normal brain, but this finding was based only on three samples and should be confirmed [130].

In conclusion, compared with healthy tissues, PTPN13 expression is decreased in all tumor types that have been studied, except for glioblastoma. As PTPN13 expression is not detectable in bone, its relative variation in Ewing sarcoma cannot be assessed. Overall, PTPN13 expression is linked to less aggressive tumors and better patient survival. PTPN13 is considered an independent biomarker of good prognosis in several solid tumor types, such as breast cancer [126], HGSOC [89,125], and HCC [90]. However, all these results are from retrospective studies, and should be confirmed in prospective studies. 

Noteworthy, many mechanistic studies that indicate a negative role of PTPN13 in FAS-induced apoptosis and suggest a pro-tumor function were performed in hematopoietic cancers. To our knowledge, no clinical study investigated the prognostic role of PTPN13 expression in these cancers. 

### 4.2. PTPN13 and Drug Sensitivity

As PTPN13 induces resistance to FAS-mediated apoptosis, several studies evaluated whether resistance to cancer treatment is associated with PTPN13 expression (see chapter 2.1). It is important to note that the results presented below were all obtained in vitro. In colon cancer cells, where FAS receptor is strongly expressed [131], a study showed that oxaliplatin induces PTPN13 expression, thereby protecting cells from apoptosis. PTPN13 silencing by siRNA combined with oxaliplatin improves sensitivity to chemotherapy by increasing FAS-induced apoptosis [55]. In line with these data, inhibition of the PTPN13/FAS interaction with the SLV peptide in PTPN13-overexpressing CD133+ colon cancer stem cells increases their sensitivity to oxaliplatin, restoring FAS-induced apoptosis [51]. 

Similar observations have been made using CML stem cells that overexpress PTPN13 and exhibit resistance to tyrosine kinase inhibitors (TKI). The combination of TKI against BCR/ABL and FAS inhibition with the SLV peptide restores sensitivity to FAS-induced apoptosis and to TKIs in such cells [61].

On the other hand, in NSCLC cell lines (SPCA1 and PC-9), PTPN13 increases the sensitivity to an anti-EGFR TKI (gefitinib). Indeed, PTPN13 inhibition by miR-26a has been associated with resistance to gefitinib in vitro and in vivo, implying in this case the ability of PTPN13 to inhibit SRC [74].

Ephrin B1 also is implicated in the response to anti-tumor treatments. Indeed, when dephosphorylated, ephrin B1 can bind to mitotic spindle microtubules, thus increasing the sensitivity to paclitaxel in head and neck squamous cell carcinoma and breast cancer cell lines [132]. Conversely, when phosphorylated, ephrin B1 is excluded from the spindle and is associated with resistance to paclitaxel in epithelial cancer cell lines. Ephrin B1 dephosphorylation is directly regulated by PTPN13, and this may explain why tumors with low PTPN13 expression are resistant to paclitaxel [132].

Another study showed that the newly identified ephrin B1/ErbB/PTPN13 signaling pathway [71] is implicated in the resistance to a monoclonal antibody against ERBB1 (cetuximab) in HNSCC. Cetuximab blocks ERBB1, but does not prevent ephrin B1 activation or ERK phosphorylation. This new signaling pathway, regulated by PTPN13, allows cells to escape treatment-induced pressure by shifting from ERBB1 to ephrinB1 signaling [71]. This suggests a potential role for the ephrinB1/ErbB/PTPN13 axis in HNSCC resistance to some EGF-R inhibitors. 

A recent study demonstrated PTPN13 involvement in cisplatin sensitivity of HNSCC cell lines (WSU-HN6 and CAL-27) where cancer-derived IgG inhibition upregulates PTPN13, resulting in the inhibition of the SRC/PKD1/AKT pathway [133].

In summary, depending on the tumor type, PTPN13 may regulate resistance to anti-cancer therapies through its anti-apoptotic role via the FAS pathway, or through regulation of secondary EGFR pathways (ephrinB1/ErbB and SRC/PKD1/AKT), or through an indirect action on microtubules mediated by ephrinB1.

### 4.3. PTPN13 Gene Alterations 

#### 4.3.1. Loss of Heterozygosity (LOH)

PTPN13 gene deletion has been observed in 37% of NSCLC [67], with higher prevalence in metastatic than non-metastatic lung cancer [134]. Similarly, PTPN13 LOH has been reported in 67% of HGSOC [135], and PTPN13 bi-allelic loss in 26% of NSCLC samples [67].

#### 4.3.2. Single Nucleotide Polymorphisms (SNP)

Besides LOH, other genetic variations may be relevant to the residual allele, particularly SNPs [136].

The Y2081D Tyr2081Asp (T > G), rs989902 (rs for SNP reference), in exon 39, near the PTPN13 phosphatase domain [137], is associated with colorectal cancer in Polish patients (relative risk compared to the “wild-type” genotype: 2.087) [138], and with HNSCC in American patients (Odds Ratio, OR, =1.26) [139]. A meta-analysis of data from Caucasian and Asian patients confirmed its association with HNSCC (OR = 1.23), but a protective role was attributed to PTPN13 in colorectal cancer (OR = 0.51). These apparently contradictory results might be explained by the different populations. In breast cancer, this meta-analysis found a protective role for this SNP (OR = 0.63), as well as a Chinese study on the C/A and G/C genotypes of this SNP (OR = 0.63 and OR= 0.66) [140].

The I1522M Ile1522Met (A > G), rs2230600, in exon 29 within the third PTPN13 PDZ domain, has been associated with HNSCC occurrence (OR = 1.89) [137]. Mita et al. [137] found an increased risk of lung, head, and neck cancer, and colorectal cancer when at least one of these two SNPs is present (adjusted OR = 3.36–13.75). Conversely, esophageal cancer was not associated with any of them.

The effects of these two SNPs on PTPN13 protein expression remain unknown. Conversely, the E2455D and Y2260WX SNPs in the catalytic domain of PTPN13 that have been identified in colorectal cancer induce a loss of 50 to almost 100%, respectively, of the phosphatase activity [37].

In addition, the nonsense Tyr1758*** (T > A) and missense Glu745Gln mutations have been found in hepatitis B virus-induced HCC samples, and the L1424P false-sense mutation, which is located in a protein-interacting PDZ domain (genomic position 87687597), may affect PTPN13 function. Around 6% of TCGA gastric cancer samples harbor PTPN13 mutations that have been associated with poor prognosis [141].

In 262 patients with familial lung cancer, a non-synonymous PTPN13 exon variant (rs115836094) at 4q21.3-28.3 could be involved in carcinogenesis [142]. Moreover, 8% of NSCLC samples harbored false-sense PTPN13 mutations with unknown functions (e.g., A808C leading to N270H in exon 7, and G1925A, leading to R482Q in exon 10) [50]. Conversely, sequencing of PTPN13 exon 7, which could be involved in the regulation of FAS-related apoptosis, in 103 colorectal cancer samples did not reveal any mutation [143]. 

Overall, approximately 7–8% of lung cancer [67,128] and 20% of HPV-negative HNSCC [144] samples harbor PTPN13 mutations. A mutational analysis of data from a tyrosine phosphatome-wide study of 157 CRC samples showed that PTPN13 is the second most commonly mutated phosphatase in these cancers (n = 15 tumors with a mutation; 9% of the entire sample) [145].

PTPN13 genetic polymorphisms need to be better investigated. Nevertheless, all analyzed mutations have an inhibitory effect on PTPN13 activity, and their presence appears to be associated with poor prognosis in lung cancer (*p* = 0.02) [128] and possibly in gastric cancer [141].

### 4.4. Bio-Informatic Analysis of the PTPN13 Gene Regulatory Network 

In 2015, Yu et al. [146] created a gene regulatory network using an innovative modeling technique (statistical completion of a partially identified graph), that is based on classical statistical data and that integrates already known mechanistic data, such as protein interactions and transcription factor binding sites. They identified PTPN13 as a new lung cancer pivotal gene and validated its prognostic importance retrospectively in four independent lung cancer datasets (n = 529 patients). The decrease in PTPN13 expression was associated with poorer prognosis.

Finally, genetic studies support the notion of PTPN13 loss of expression/function in cancer, through LOH and/or SNPs. The previously described epigenetic mechanisms have the same negative consequences on its expression.

Altogether, these data bring weight to the hypothesis of a primary tumor suppressor role for the PTPN13 phosphatase.

## 5. Conclusions

The work of the last 25 years on PTPN13 reveals that this phosphatase is involved in many physiological mechanisms, with a variable importance depending on the cell type.

In solid tumors, PTPN13 tumor suppressor role, via inhibition of pathways involved in cell proliferation and migration, seems to be confirmed by clinical studies. Conversely, its potential pro-oncogenic role in hematologic malignancies, via resistance to FAS-induced apoptosis, needs to be investigated at the clinical level. 

Interestingly, recent studies on PTPN13 highlight its function as a stabilizer of epithelial cell junctions, a role that in epithelial tissues, places PTPN13 at the interface between cell migration and cell death and should be better assessed in the future [30,83,84].

Finally, although PTPN13 is primarily studied in cancer, its roles in multiple signaling pathways suggest its implication also in other pathologies, for instance, neurodegenerative diseases where recent studies pointed to a potential role in tau phosphorylation [147].

## Figures and Tables

**Figure 1 biomolecules-10-01659-f001:**
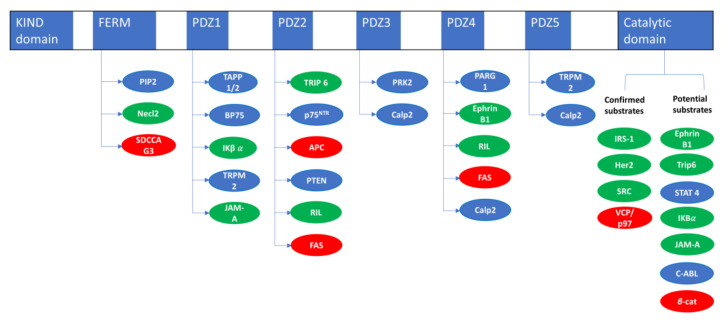
PTPN13 structure, interactions, and substrates. Interactors or substrates in green were involved in PTPN13 tumor suppressor role. Interactors or substrates in red were involved in PTPN13 pro-tumoral role. KIND domain: kinase non-catalytic C-lobe domain (unknow function), FERM: 4.1/Ezrin/radixin/moesin domain (protein/protein and protein/plasma membrane interaction), PDZ domains: PSD95/Dlg1/Zo-1 domain (protein/protein interaction domain), Calp2: Calpain-2, β-cat: β-catenin.

**Figure 2 biomolecules-10-01659-f002:**
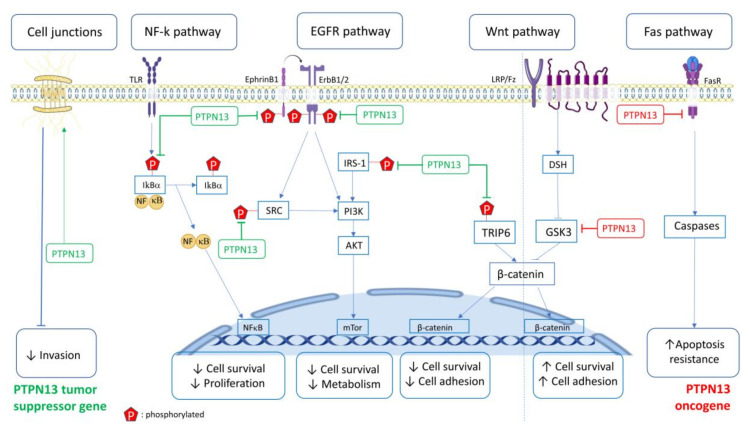
PTPN13 is involved in multiple signaling pathways. PTPN13 in green: Tumor suppressor role; PTPN13 in red: Pro-tumoral role. Arrow pointing to P: PTPN13 effects mediated by its phosphatase activity. TLR: Toll-like receptor, LRP/Fz: lipoprotein receptor-related protein/Frizzled.

**Table 1 biomolecules-10-01659-t001:** PTPN13 interacting proteins.

PTPN13 Interacting Proteins		
Name	Interacting Domain	Reference
Necl2	FERM	[18]
SDCCAG3/ENTR1	FERM	[15]
TAPP 1/2	PDZ1	[14]
BP75	PDZ1	[19]
IκBα	PDZ1	[20]
TRPM2	PDZ15	[21]
JAM-A	PDZ-1	[22]
TRIP6/ZRP1	PDZ2	[23]
P75^NTR^	PDZ2	[24]
APC	PDZ2	[25]
RIL	PDZ2-4	[26]
FAS	PDZ2-4	[27]
PTEN	PDZ2	[28]
PRK2	PDZ3	[29]
Calp2	PDZ3-4-5	[30]
PARG1	PDZ4	[31]
EphrinB1	PDZ4	[32]

**Table 2 biomolecules-10-01659-t002:** PTPN13 Susbstrates.

PTPN13 Substrates Evidences			
Name	Dephosphorylation	Substrate Trapping	Reference
IRS1	In vitro/in cellulo	In vitro/in cellulo	[36]
HER2	In cellulo	In cellulo	[37]
SRC	In vitro/in cellulo	In vitro/in cellulo	[38]
VCP/P97	In vitro/in cellulo	In vitro/in cellulo	[39]
EphrinB1	In vitro/in cellulo		[40]
Trip6	In vitro/in cellulo		[41]
STAT 4	In vitro/in cellulo		[33]
IκBα	In vitro/in cellulo		[42]
JAM-A	In vitro		[22]
C-ABL	In cellulo		[30]
β-catenin	In cellulo		[43]

**Table 3 biomolecules-10-01659-t003:** Transcription factors and co-repressors regulating PTPN13.

Transcription Factors and Co-Repressors Regulating PTPN13
Name	Expression/Tumor Type	Effect	Reference
STAT3/HDAC5	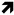 Lung cancers	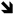 PTPN13	[101]
SMYD2	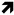 Polycystic kidney disease 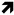 Breast cancer cell lines	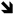 PTPN13 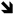 PTPN13	[103][102]
HBx/DNMT3A	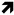 Liver cancer	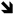 PTPN13	[104]
ICSBP	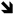 Chronic myeloid leukemia	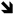 PTPN13	[106]
EWS-FLI1	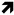 Ewing’s sarcomas	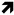 PTPN13	[108]
Hox-C8	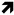 Cell differentiation	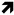 PTPN13	[109]

**Table 4 biomolecules-10-01659-t004:** miRNA targeting PTPN13.

miRNA Targeting PTPN13			
Name	Tumor Type	miRNA Expression	Reference
miR-26-a	Lung cancers	Overexpression	[74]
miR30-e	Lung cancers	Overexpression	[75]
miR-200c	Various cancers	Loss of expression	[56]
miR-185	Bladder and Kidney Cancers	Overexpression	[121]

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
