# Peer review of "Dual Role of the PTPN13 Tyrosine Phosphatase in Cancer"

_biomolecules, 2020, doi:10.3390/biom10121659_

Round 1
Reviewer 1 Report
This is an informative and nicely written review paper about the role of the phosphatase PTPN13 in cancer. The text is straight forward and the illustrations have been carefully prepared. Together, this review will be of interest to both the PTPN13 specialists as well as the neophytes.
The following recent paper should be added and discussed:
Yan Y, Huang P, Mao K, He C, Xu Q, Zhang M, Liu H, Zhou Z, Zhou Q, Zhou Q, Ou B, Liu Q, Lin J, Chen R, Wang J, Zhang J, Xiao Z. Anti-oncogene PTPN13 inactivation by hepatitis B virus X protein counteracts IGF2BP1 to promote hepatocellular carcinoma progression. Oncogene. 2020 Oct 13. doi: 10.1038/s41388-020-01498-3. Epub ahead of print. PMID: 33051595.
Author Response
The following recent paper should be added and discussed:
Yan Y, Huang P, Mao K, He C, Xu Q, Zhang M, Liu H, Zhou Z, Zhou Q, Zhou Q, Ou B, Liu Q, Lin J, Chen R, Wang J, Zhang J, Xiao Z. Anti-oncogene PTPN13 inactivation by hepatitis B virus X protein counteracts IGF2BP1 to promote hepatocellular carcinoma progression. Oncogene. 2020 Oct 13. doi: 10.1038/s41388-020-01498-3. Epub ahead of print. PubMed ID 33051595.
This very recent reference has now been added and is discussed in chapter 3.1.1 (lines 305-311) as following:
“More recently, Yan et al. demonstrated that, in HCC, the hepatitis B virus X protein regulates PTPN13 expression via the DNA methyltransferase 3A which binds to the PTPN13 promoter and induces the hypermethylation of its CpG islands. This loss of PTPN13 leads to an increase in c-Myc levels and signaling through the loss of its competitive interaction with IGF2PB1 that protects c-Myc mRNA from degradation. ”
Reviewer 2 Report
Mcheik et al. present a comprehensive review on the role of the protein tyrosine phosphatase PTPN13 in cancer. The manuscript is well written and with an appropriate and up-to-date coverage of the subject, and will be of interest for researchers in the field.
Below are a few comments regarding some aspects of the manuscript that could be improved, as well as regarding some small corrections needed.
- The dual role of PTPN13 in cancer forms part of the title and it is mentioned along the manuscript as a major point of it, especially to make a distinction between solid tumors (potential tumor suppressor role of PTPN13) and hematologic tumors (potential oncogenic role of PTPN13). This concept could be reinforced by including it with a bit of detail in the legend of Figure 2. It is also suggested to make two colors for PTPN13 depiction in all the Figure 2: one color accounting for tumor suppressor functions and another color accounting for oncogenic functions, in accordance with the color code already used in the bottom of the figure. Following this rational, it is suggested to use this two color code in the depiction of PTPN13 domains, interactors, and substrates in Figure 1. Finally, the legend of Figure 2 should also explain the distinction between PTPN13 effects mediated by its phosphatase activity and those mediated by its scaffolding properties.
- Potential dephosphorylation of targets by PTPN13 shown in Figure 2 should match the substrates listed in Figure 1: are PI3K and AKT potential or confirmed substrates of PTPN13? (they are not shown in Figure 1); IRS-1 is listed as a confirmed PTPN13 substrate but it is not shown in Figure 2; IKBa in Figure 1 corresponds to IFKB in Figure 2?
- Some words in Figure 1 and in Figure 2 are very small and difficult to read. This could be amended.
- To facilitate the finding of relevant information by the reader, it is suggested to add some Tables to the manuscript that summarize the PTPN13 binding partners and substrates together with the source references. The same can be applied to summarize the PTPN13 transcription factors and miRNAs.
- Lines 210 and 211 should be together, with a period after ref 76 (instead of being in two paragraphs). In line 212-213, and in line 413, there are indications to “chapters” that do not exist in the manuscript.
- It is suggested to modify or remove some terms or abbreviations that, in opinion of this reviewer, are rarely used in the literature: “(post)transcriptional”, “(epi)genetics” should be changed by “post-transcriptional”, “epigenetics”; “TR”, “PTR” could be removed.
- Line 364 should read “Post-translational regulation”. This section (3.3) could incorporate a comment on the large number of PTPN13 residues post-translationally modified found in proteomic studies, as illustrated in www.phosphosite.org.
- Line 472-473: change “false-sense” by “missense”.
- Lines 501-504, and lines 517-518: the sentences “Authors should discuss…also be highlighted” and “This section … Article” do not belong here and need to be removed.
- In line 509, it is suggested to write “Conversely, its potential pro-oncogenic role…”
Author Response
- The dual role of PTPN13 in cancer forms part of the title and it is mentioned along the manuscript as a major point of it, especially to make a distinction between solid tumors (potential tumor suppressor role of PTPN13) and hematologic tumors (potential oncogenic role of PTPN13). This concept could be reinforced by including it with a bit of detail in the legend of Figure 2. It is also suggested to make two colors for PTPN13 depiction in all the Figure 2: one color accounting for tumor suppressor functions and another color accounting for oncogenic functions, in accordance with the color code already used in the bottom of the figure. Following this rational, it is suggested to use this two color code in the depiction of PTPN13 domains, interactors, and substrates in Figure 1. Finally, the legend of Figure 2 should also explain the distinction between PTPN13 effects mediated by its phosphatase activity and those mediated by its scaffolding properties.
Both figures1 and 2 have been modified as requested.
- Potential dephosphorylation of targets by PTPN13 shown in Figure 2 should match the substrates listed in Figure 1: are PI3K and AKT potential or confirmed substrates of PTPN13? (they are not shown in Figure 1); IRS-1 is listed as a confirmed PTPN13 substrate but it is not shown in Figure 2; IKBa in Figure 1 corresponds to IFKB in Figure 2?
We agree that during the drafting of the schemes some discrepancies between the 2 figures have been introduced. Figure 2 has now been amended by expanding the signaling pathways to indicate the direct PTPN13 substrates.
- Some words in Figure 1 and in Figure 2 are very small and difficult to read. This could be amended.
The small words difficult to read have now been enlarged.
- To facilitate the finding of relevant information by the reader, it is suggested to add some Tables to the manuscript that summarize the PTPN13 binding partners and substrates together with the source references. The same can be applied to summarize the PTPN13 transcription factors and miRNAs.
Referenced tables have now been added to summarize the PTPN13 binding partners and substrates (Table 1 and 2) and PTPN13 transcription factors and miRNAs (Table 3 and 4) and referred to in the manuscript text.
- Lines 210 and 211 should be together, with a period after ref 76 (instead of being in two paragraphs). In line 212-213, and in line 413, there are indications to “chapters” that do not exist in the manuscript.
The two lines are now combined ( lines 222-225) and the chapter numbers have been edited .
- It is suggested to modify or remove some terms or abbreviations that, in opinion of this reviewer, are rarely used in the literature: “(post)transcriptional”, “(epi)genetics” should be changed by “post-transcriptional”, “epigenetics”; “TR”, “PTR” could be removed.
The terms have been modified and the abbreviations have been replaced by the corresponding correct names .
- Line 364 should read “Post-translational regulation”. This section (3.3) could incorporate a comment on the large number of PTPN13 residues post-translationally modified found in proteomic studies, as illustrated in www.phosphosite.org.
We have added a sentence to indicate that there are many PTPN13 post-translational modifications, but that their role has unfortunately not yet been elucidated. (Lines 391-393)
- Line 472-473: change “false-sense” by “missense”.
The correction has been made.
- Lines 501-504, and lines 517-518: the sentences “Authors should discuss…also be highlighted” and “This section … Article” do not belong here and need to be removed.
The authors apologize for this editing error that occurred during the conversion in the journal’s format. The corresponding text has now been removed (line526).
- In line 509, it is suggested to write “Conversely, its potential pro-oncogenic role…”
The authors agree with referee’s suggestion and modified the text accordingly (lines 531).
Reviewer 3 Report
The review by Mcheik et al. is summarising the cell biology and relevance to cancer of the protein tyrosine phosphatase PTPN13. PTPN13’s cell biology is complex, and the protein appears to act in a number of different contexts. Thus, it is indeed an important task and service to the research community to summarise and importantly to structure our current knowledge we have about this protein.
The review constitutes largely of brief summaries of individual papers that are categorised under distinct subheadings focusing on specific aspects of PTPN13’s (cancer) cell biology. The subheadings are well chosen. However, the overall approach and in parts lack of connecting findings (synthesis) from different papers into a more coherent picture makes it very difficult to grasp overarching concepts from this review. Unfortunately, this makes the review also quite difficult to read as the information density and level of detail is high. On the other hand, this review is very useful to a reader to find relevant literature for a specific PTPN13 topic and to get a first brief overview of the content of such papers. This in itself is a merit. Furthermore, the two figures are integrating relevant interactions and signalling pathways well.
In addition, the last chapter (medical implication) performs in part more synthesis of existing clinical data and thus is in particular meaningful.
Minor points:
The authors claim that PTPN13 was cloned by 4 different groups hence 4 different names, however Sato et al. (1995) had renamed the protein due to its interaction with Fas, as far as I know it was already cloned at this point.
Line 42: Hela cells are not polarised and thus don’t have an apical domain.
Line 47: Explain ENTR1 abbreviation
Line 72: the claim interaction with PDZ3. There is some confusion in the literature as Sato et al. proposed six PDZ domains for FAP-1, however most publications report 5 PDZ domains. This leads sometimes to confusion with the numbering (PDZ3 Sato actually being PDZ2). I suggest to double check the statement.
Check references for duplications: Citation 15 and 35 are identical
Author Response
The review constitutes largely of brief summaries of individual papers that are categorised under distinct subheadings focusing on specific aspects of PTPN13’s (cancer) cell biology. The subheadings are well chosen. However, the overall approach and in parts lack of connecting findings (synthesis) from different papers into a more coherent picture makes it very difficult to grasp overarching concepts from this review. Unfortunately, this makes the review also quite difficult to read as the information density and level of detail is high. On the other hand, this review is very useful to a reader to find relevant literature for a specific PTPN13 topic and to get a first brief overview of the content of such papers. This in itself is a merit. Furthermore, the two figures are integrating relevant interactions and signalling pathways well.
In addition, the last chapter (medical implication) performs in part more synthesis of existing clinical data and thus is in particular meaningful.
The authors agree that the PTPN13 dual effect on cancer aggressiveness makes it difficult to sum up and summarize all PTPN13 activities in a clearcut way. We therefore opted to describe all known PTPN13 activities by classifying them by signaling pathway and make a brief synthesis at the end of each chapter. The two figures are integrating relevant interactions and signaling pathways, and the modifications made in the 2 figures are intended to simplify their reading and to better understand the dual PTPN13 activities.
Minor points:
The authors claim that PTPN13 was cloned by 4 different groups hence 4 different names, however Sato et al. (1995) had renamed the protein due to its interaction with Fas, as far as I know it was already cloned at this point.
Indeed, Sato et al. cloned a short cDNA with high homology to PTP-BAS and renamed it FAP-1. The manuscript has been edited in this sense “PTPN13 was first named PTP-BAS[1], hPTP1E[2]] and PTP-L1[3] since it was cloned by three different groups, then renamed FAS-associated phosphatase 1 (FAP-1) by Sato et al because of its interaction with Fas[4]”. (lines 30-31)
Line 42: Hela cells are not polarised and thus don’t have an apical domain.
The authors agree that Hela cells are not clearly polarized. In our experiments PTPN13 was localized to the plasma membrane on the upper side of the Hela cells, but it is indeed not accurate to call it an apical domain. We therefore removed the reference to apical domain and stated “the FERM domain is necessary and sufficient to direct the enzyme to the membrane”. (lines 43-44)
Line 47: Explain ENTR1 abbreviation
The abbreviation has now been explained: endosome-associated trafficking regulator 1 (line 59)
Line 72: the claim interaction with PDZ3. There is some confusion in the literature as Sato et al. proposed six PDZ domains for FAP-1, however most publications report 5 PDZ domains. This leads sometimes to confusion with the numbering (PDZ3 Sato actually being PDZ2). I suggest to double check the statement.
The authors agree that they have been misled by the numbering of Sato. We rectified this error and added the interaction with PDZ4 and added the reference by Saras et al (J. Biol. Chem 1997) that best describes these interactions. “that the second and fourth PDZ domain of PTPN13 directly interact with FAS C-terminus”
Check references for duplications: Citation 15 and 35 are identical
The references were changed accordingly.
Round 2
Reviewer 2 Report
The authors have addressed satisfactorily all suggestions by this reviewer